# Prenatal cfDNA Screening for Emanuel Syndrome and Other Unbalanced Products of Conception in Carriers of the Recurrent Balanced Translocation t(11;22): One Laboratory’s Retrospective Experience

**DOI:** 10.3390/genes14101924

**Published:** 2023-10-10

**Authors:** Erica Soster, Brittany Dyr, Samantha Caldwell, Amanda Sussman, Hany Magharyous

**Affiliations:** 1Labcorp, La Jolla, San Diego, 92121 CA, USA; dyrb@labcorp.com (B.D.); caldws2@labcorp.com (S.C.); magharh@labcorp.com (H.M.); 2Labcorp, Research Triangle Park, Durham, 27709 NC, USA; sussmaa@labcorp.com

**Keywords:** cell-free DNA (cfDNA), noninvasive prenatal testing (NIPT), prenatal screening, prenatal diagnosis, Emanuel syndrome, translocation

## Abstract

Prenatal cell-free DNA screening (cfDNA) can identify fetal chromosome abnormalities beyond common trisomies. Emanuel syndrome (ES), caused by an unbalanced translocation between chromosomes 11 and 22, has lacked a reliable prenatal screening option for families with a carrier parent. A cohort of cases (*n* = 46) sent for cfDNA screening with indications and/or results related to ES was queried; diagnostic testing and pregnancy outcomes were requested and analyzed. No discordant results were reported or suspected; there were ten true positives with diagnostic confirmation, six likely concordant positives based on known translocations and consistent cfDNA data, and twenty-six true negatives, by diagnostic testing or birth outcomes. For cases with parental testing, all affected ES cases had maternal translocation carriers. Expanded cfDNA may provide reassurance for t(11;22) carriers with screen negative results, and screen positive results appear to reflect a likely affected fetus, especially with a known maternal translocation. Current society guidelines support the use of expanded cfDNA screening in specific circumstances, such as for translocation carriers, with appropriate counseling. Diagnostic testing is recommended for prenatal diagnosis of ES and other chromosome abnormalities in pregnancy. To our knowledge, this cohort is the largest published group of cases with prenatal screening for carriers of t(11;22).

## 1. Introduction

Emanuel syndrome (ES) [OMIM #609029] is a rare condition caused by a supernumerary derivative chromosome 22. The condition, named after Dr. Beverly Emanuel in 2004, has had many different names. A parent group chose the name to honor Dr. Emanuel’s contributions to the understanding of the disorder and to make it easier for families to connect under a single disorder name, as well as to delineate the condition from other genetic disorders involving the same region of chromosome 22 [1,2]. Typically, ES occurs after 3:1 meiotic segregation during gamete formation in a parent carrying a balanced reciprocal translocation between chromosomes 11 and 22: t(11;22)(q23;q11) [2,3,4], henceforth abbreviated as t(11;22). A single case report notes a de novo origin of the unbalanced translocation product; the vast majority of cases are inherited [5]. The true prevalence of ES is unknown, but was estimated in one study to be ~1 in 110,000 [6]. The same study estimated the prevalence of t(11;22) carriers to be ~1 in 16,000; this translocation is the most common recurrent non-Robertsonian translocation [5,7].

The typical karyotypes for ES are either 47,XX,+der(22)t(11;22)(q23;q11) in females or 47,XY,+der(22)t(11;22)(q23;q11) in males. The unbalanced der(22) associated with ES is comprised of a duplication of 22q that is approximately 3–4 Mb in size and a duplication of chromosome 11q that is approximately 18 Mb in size [8,9]. Usually, unbalanced products can be detected by routine G-banded karyotype or FISH. However, the smaller size of the chromosome 22 rearrangement and banding pattern makes detection of the balanced t(11;22) challenging by routine G-banded analysis. Chromosomal microarray can detect the unbalanced segments on chromosomes 11 and 22 but cannot confirm the structure. FISH may also be useful as there are multiple commercial probes available for 22q and subtelomeric probes can be used for 11q. The translocation, whether balanced or unbalanced, is typically inherited from a carrier parent; however, for affected individuals, the translocation is predominantly maternally inherited [3,5,7]. Thus, the risk for an affected child (or recurrence risk) is highest if the mother is a carrier [2,3,5,7], although paternal transmission has also been reported [4,10]. There is limited information regarding the segregation of t(11;22), particularly in female carriers. The rate of 3:1 segregation appears to be higher in maternal carriers, perhaps as a consequence of meiotic arrest during female gametogenesis [3,4,7]. Male carriers appear to generate all forms of unbalanced segregations, including those forms that are unviable, which likely contribute to the increased rate of failed implantation and early pregnancy loss in these families [4,7,10]. This may explain the predominant maternal inheritance in affected cases.

For carriers of t(11;22) the potential products for a fetus/child are: euploid, no translocation; euploid but carrying the balanced translocation (and phenotypically normal); miscarriage due to the supernumerary der(22) or another unbalanced byproduct of the translocation; or an individual affected with ES due to a supernumerary der(22). The exact risk of each outcome is not known, but as noted above, the risk of transmitting a supernumerary der(22) is highest if the mother carries the translocation. One study estimates the risk for a live-born child with ES to be 1.8–5.6% and the risk for early pregnancy loss due to supernumerary der(22) or another unbalanced segregation product to be 23–37% [3,5,7].

Counseling, especially prenatally, can be challenging as the outcomes can range from early pregnancy loss to neonatal death to long-term survival with significant developmental delays and medical complications. Commonly reported features of ES include growth restriction (prenatally and postnatally), dysmorphic features, cardiac defects, cleft palate, renal anomalies, diaphragmatic hernia, hypotonia, severe developmental delays, musculoskeletal abnormalities, brain anomalies (including Dandy–Walker syndrome), and microcephaly [2,5]. Dysmorphic facial features may be recognizable and include a prominent forehead, microretrognathia, ear malformations, deeply set and round eyes, a broad, depressed nasal bridge, and a long philtrum [5]. Mortality is often related to the major structural anomalies and is highest in the months after birth [5]. Only some of these findings, such as growth restriction, cardiac defects, diaphragmatic hernia, and brain anomalies, may be ascertained prenatally by routine ultrasound. In one study surveying families of affected individuals, the majority of families (81%) reported no pregnancy complications and approximately half (48%) received their child’s diagnosis within a month of their birth [2]. The same study reported that only one family received a prenatal diagnosis and many families reported extended neonatal hospital stays due to medical complications [2]. There are no published care or management guidelines available for individuals with ES, but individuals with this condition will require care from a multidisciplinary team; in some cases, palliative care may be considered [5].

As a relatively common translocation, families who carry t(11;22) are presenting for reproductive care. Limited data exist on prenatal screening and the diagnosis of ES; biochemical screening for ES is not well studied; but, in one study, it is evident that biochemical screening is unlikely to be a sensitive (or specific) screen for the condition, as only one of eight cases across two cohorts had an abnormal biochemical screen [9,11]. Another study noted that only 16% of cases in a series of patients with ES had anomalies detected prenatally on ultrasound [2]. Luo et al. report on two cases with ES identified by prenatal cfDNA screening and confirmed by diagnostic testing [8]. To our knowledge, Luo’s publication is the only report specifically on the use of prenatal cfDNA screening for ES, although there are cases included in the cohort by Flowers et al. [12].

Although prenatal cfDNA screening (also called noninvasive prenatal testing [NIPT] or noninvasive prenatal screening [NIPS]) typically focuses on the common viable aneuploidies (trisomies of chromosomes 21, 18, and 13), expanded cfDNA is available for a broader range of chromosome abnormalities, including large CNVs and specific microdeletion syndromes. Professional societies currently do not recommend routine screening via expanded cfDNA, yet there is some support for using expanded cfDNA in specific situations. In their updated prenatal cfDNA screening guidelines, the American College of Medical Genetics (ACMG) [13] noted ‘[…] there will be families for whom NIPS for CNVs could be offered based on the pregnancy or family history’. They also emphasize the importance of pretest counseling for these situations. The International Society for Prenatal Diagnosis, regarding screening for CNVs, states [14]: ‘In selected circumstances, however, it [cfDNA screening] may be of clinical utility, for example for carriers of balanced reciprocal translocations’.

This case series aims to add to the existing literature on prenatal ascertainment of ES, as well as provide preliminary evidence for the role of prenatal cfDNA screening in carriers of t(11;22), particularly in regard to ES.

## 2. Materials and Methods

Maternal blood samples were submitted for prenatal cfDNA screening via either the ‘traditional’ assay (MaterniT^®^ 21 Plus with Enhanced Sequencing Series) or genome-wide cfDNA screening (MaterniT^®^ GENOME) as designated by the ordering provider. MaterniT^®^ 21 PLUS screens for trisomy of chromosomes 21, 18, and 13 with the option of expanded content (sex chromosome aneuploidies, and the Enhanced Sequencing Series which includes trisomies 16 and 22, as well as microdeletions associated with 1p36 deletion, Wolf–Hirschhorn, Cri-du-chat, Langer–Giedion, Jacobsen, Prader–Willi, Angelman, and DiGeorge syndromes). MaterniT^®^ GENOME screens for aneuploidy of any chromosome, copy number variants (CNVs) ≥7 Mb in size, as well as the select microdeletions <7 Mb in size mentioned above. Of note, duplications of 22q less than 7 Mb are not validated on either assay but may be reported or noted in the laboratory director’s comments if the duplication is seen in conjunction with a CNV larger than 7 Mb.

For the ‘traditional’ cfDNA assay, although the involved CNVs are outside of the scope of testing, a verbal review of the masked sequencing data was requested by the ordering provider in each case. Laboratory genetic coordinators would then discuss the sequencing data with the provider. Redraw and/or resequencing of the sample on the genome-wide assay was offered as appropriate, but declined by the provider(s). Of note, the genome-wide assay is not validated in twin pregnancies; when ordered in error on a twin pregnancy, redraw and/or resequencing on the ‘traditional’ cfDNA assay was offered. If the offer was declined, the report was amended to state: “The report dated [date] is amended to change the clinically reported gestational status from singleton to twins. MaterniT Genome is not validated in multifetal pregnancies”.

Blood samples were subjected to DNA extraction, library preparation, and genome-wide massively parallel sequencing (MPS), as previously described [15], with detection of subchromosomal CNVs as described by Zhao et al. [16]. For cases submitted for genome-wide analysis, sequencing data were analyzed using a proprietary algorithm (using Z-score along with other metrics) to detect aneuploidies and other subchromosomal events as described by Lefkowitz et al. [17]. Fetal fraction was estimated as described previously [18] and each sample was required to meet a sample-specific fetal fraction threshold utilizing signal-to-noise ratios (SNR) to be considered reportable [19]. Minimum fetal fraction/SNR is approximately twice as high for twin pregnancies. Published non-reportable rates for the ‘traditional’ cfDNA assay are ~1% [20,21] and are slightly higher (~4%) [22] for the genome-wide assay given the more stringent SNR requirements to reliably call genome-wide CNVs. Pre- and post-test counseling and informed consent were the responsibility of the clinicians ordering the testing.

A retrospective database search was conducted to query samples related to ES and/or the associated 11;22 translocation since the launch of the additional content in 2013 through April 2023. The database includes over 2 million prenatal cfDNA samples. An initial search was conducted of the clinical database for screen-positive cases involving complex CNVs on chromosomes 11 and 22; those cases were manually reviewed to see if the CNVs were consistent with ES and/or the associated translocation. Next, screen-negative cases were reviewed for those that were tested due to a known familial translocation and those that involved the 11;22 translocation were included in the dataset. Lastly, for patients who were identified in either of the first two groups, any additional pregnancies that were sent for cfDNA screening at the same laboratory for the same patient were reviewed for confirmation of a parental carrier of t(11;22) and were also included.

Outcome information was obtained from two sources. First, both obstetric and diagnostic outcome information was collected from the ordering provider, when available. Second, cfDNA results were cross-referenced with diagnostic results (FISH, microarray, and karyotype) submitted to Labcorp from chorionic villus, amniocentesis, neonatal and/or parental peripheral blood, and products of conception specimens. Collection of outcomes and the process of consolidation and comparison of data across the datasets (cfDNA results and diagnostic results) was approved by Aspire IRB under clinical protocol SCMM-RND-402. Informed consent was not required as Aspire IRB declared that this research meets the requirements for a waiver of consent under 45 CFR 46 116(f)[2018 Requirements].

Positive cases were considered concordant (true positive) if diagnostic testing confirmed the cfDNA finding(s). Positive cases were deemed likely concordant (likely positive) if no diagnostic testing was performed but the cfDNA findings were consistent with a known maternal or paternal translocation. Consistent clinical features (such as ultrasound anomalies or pregnancy loss) were considered supportive of the ‘likely positive’ assignment. Negative cases were considered concordant (true negative) if either diagnostic testing was performed and was negative or if a healthy livebirth was reported.

Study data were statistically described using counts, rates, and measures of central tendency.

## 3. Results

### 3.1. Study Cohort

There were a total of 46 samples from 32 unique patients that met the inclusion criteria. There were twenty-two patients with one pregnancy, seven patients with two pregnancies, two patients with three pregnancies, and one patient with four pregnancies. There were three cases of twin pregnancies; the remainder were singleton pregnancies. Table 1 summarizes the relevant details from the patients and cases.

In 42 cases, a genome-wide cfDNA assay was ordered, while in 4 cases, the ‘traditional’ cfDNA assay was ordered. For the four cases involving the ‘traditional’ cfDNA assay, three of the cases reported a history of a known familial translocation, and at provider request a verbal review of the data for chromosomes 11 and 22 was relayed by the laboratory to the clinician. All three of those cases were verbally noted to have reassuring sequencing data for chromosomes 11 and 22, with the caveat that the ‘traditional’ assay was not validated for these events. For the final case, the provider requested a verbal review of the data in response to ultrasound findings as a ‘reflex’ to the expanded cfDNA assay was being considered. However, the presence of the ultrasound findings combined with the verbal note of the concerning cfDNA sequencing data on chromosomes 11 and 22 prompted the patient to pursue diagnostic testing instead.

The average gestational age at testing was 12.24 weeks, while the average maternal age was 32.70 years. Figure 1 shows the indications for testing; of note, 42/46 cases (91.3%) noted a known familial translocation as either the sole indication or in combination with another indication. Of the 42 ‘known’ translocation cases, there were 9 paternal translocation cases (6 unique patients; 3 patients with 2 cases each; and 3 patients with a single case) and 33 maternal translocation cases (1 patient with 4 cases; 1 patient with 3 cases; 5 patients with 2 cases; and 16 patients with 1 case). Of the four cases without a known familial translocation prior to the first cfDNA screening test in the cohort, one of those patients (patient 24) was identified to carry out the translocation (maternal) following an affected pregnancy (Case 24-A) and went on to have two subsequent pregnancies screened (one positive and one negative).

### 3.2. cfDNA Findings

Of the 46 cases, there were 29 cases with negative cfDNA screening results and 17 cases with positive cfDNA screening results, as shown in Figure 2. Of those 17 screen-positive cases, 2 cases were positive for findings presumably unrelated to the translocation or ES (1 case of apparently full trisomy 22 and 1 case of 45,X) but both had unremarkable data for the remainder of the chromosomes, including chromosome 11. The cfDNA findings for the 17 screen-positive cases are summarized in Table 2. This leaves 15 cases which appeared to have an unbalanced translocation. One case, 2-A, had an unusual result, suggesting a large gain (116.65 Mb) of 11p15.5-q23.3 and a deletion of 22q11.2 in the DiGeorge region, with a paternal t(11;22) translocation noted. The cfDNA screening results are consistent with the breakpoints from the translocation carrier parent but are not consistent with ES; this finding likely represents the presence of an unbalanced karyotype with der(11) and a single normal chromosome 22 following adjacent 2 segregation, which is not viable. The 14 remaining screen-positive cases had results consistent with ES. All 14 cases noted a large duplication on chromosome 11q, typically in the range of approximately 18–20 Mb. Most of the cases reported a duplication on chromosome 22q, typically approximately 3 Mb. Case 8 did not report a 22q event, nor was one noted on the retrospective data review; case 15-D did not report a 22q event, but an approximately 2 Mb gain on 22q was noted on the retrospective data review. Of the 42 cases where there was a known familial translocation, 13 had screen positive cfDNA results (11 cases involving the translocation and 2 cases of unrelated aneuploidy, trisomy 22 and 45,X) and 29 had negative cfDNA results.

### 3.3. Diagnostic Testing and Pregnancy Outcome

Diagnostic testing was available for 17 cases (10 screen-positive cases and 7 screen-negative cases), which are shaded light gray in Figure 2. All 17 cases had concordant cfDNA and diagnostic testing results. Approximately half of all screen-positive cases had diagnostic testing results (10/17, 58.8% or 10/15, 66.7% if only considering screen-positive cases related to the 11;22 translocation), while relatively fewer screen-negative cases had diagnostic testing (24.14%). Of the 42 cases where there was a known familial translocation prior to cfDNA testing, 13 cases (30.95%) had diagnostic testing; 6 had positive cfDNA screening results; and 7 had negative cfDNA screening results.

The pregnancy outcome was available for 30 cases: 21 screen-negative cases and 9 screen-positive cases (one of which was the trisomy 22 case). Of the nine screen-positive cases, three cases ended in a spontaneous pregnancy loss (one was the trisomy 22 case), four opted for elective terminations (two confirmed on diagnostic testing and two with no further testing), and two affected pregnancies (confirmed by diagnostic testing) resulted in a live birth of an affected child with ES. All four termination cases included ultrasound anomalies. Only one of the miscarriage cases had diagnostic testing confirming a pregnancy with ES. Of the 21 screen-negative cases, 20 resulted in healthy livebirths, with the remaining case resulting in the birth of a child with a Mendelian genetic disorder unrelated to the translocation (euploid on amniocentesis), diagnosed by whole exome sequencing. None of the 20 healthy livebirths had confirmatory genetic testing.

Diagnostic testing and/or pregnancy outcome were available for 91.3% (42/46) cases; if likely positive cases are excluded, 76.1% (35/46) cases could be reasonably designated as true negative or true positive based on the diagnostic testing results and/or birth outcome. Overall, between the reported diagnostic testing and pregnancy outcomes, no discordant results were reported or suspected; there were ten true positives confirmed by diagnostic testing, six likely concordant positives based on the known parental translocation and consistent cfDNA data, and twenty-six true negatives, by either diagnostic testing or reported normal birth outcomes. There were three cases with no diagnostic testing which were lost to follow-up (one of which was the positive cfDNA for 45,X).

For the 34 cases confirmed to be maternal translocation carriers, the diagnostic outcome was available in 31 cases. There were seven true positive cases (24.1%), five likely positive cases (17.2%), and nineteen true negative cases (58.6%). Three of the remaining cases were lost to follow-up.

For the nine cases confirmed to be paternal translocation carriers, there was one likely concordant atypical positive case (Case 2-A, above), while the remaining eight cases were all true negatives. Three of those true negatives included a fetus who carried a balanced (11;22) translocation (two singleton pregnancies and one presumably dizygotic twin pregnancy with one balanced carrier fetus and one 46,XY fetus). Figure 3 shows the outcomes grouped by known versus unknown translocation carriers.

Details regarding ultrasound findings in screen-positive cases are limited and noted in Table 1, but the following ultrasound findings were noted in three screen-positive cases each: diaphragmatic hernia, Dandy–Walker syndrome, micrognathia/recessed chin, and prenatal growth restriction.

## 4. Discussion

### 4.1. Overview and Comparison to Literature

To our knowledge, this cohort is the largest published group of cases with prenatal screening for carriers of t(11;22). Two publications note a small number cases of ES identified via prenatal cfDNA screening [8,12], while prenatal serum biochemical screening has only been mentioned in a handful of publications and does not appear to have meaningful utility in screening for this condition [9,11]. Ultrasound findings for ES are non-specific and overlap with a variety of other genetic syndromes and may not be present in all cases of ES [2,5,23]; therefore, the absence of ultrasound findings may not be reassuring.

Based on our cohort, it appears that expanded cfDNA can provide reasonable reassurance for t(11;22) carriers in the event of screen negative results and that screen positive results are associated with a high likelihood of a fetus affected with ES, especially when it is a known maternal translocation. Undoubtedly, diagnostic testing will continue to be the gold standard for prenatal diagnosis of ES and other chromosome abnormalities in pregnancy. However, families that are known to carry translocations may have poor obstetric history with multiple pregnancy or perinatal losses and may be especially averse to a diagnostic procedure in a pregnancy that has continued past the late first trimester [24]. Many of the patients in our cohort were noted to have a history of pregnancy losses. Preimplantation genetic testing and/or IVF is also an option for families with known translocations; there were three cases in our cohort where PGT for the structural rearrangement (PGT-SR) was performed and euploid/balanced embryos were transferred and an additional case used an egg donor for conception in the presence of a known maternal translocation. All of these cases were true negatives.

The language of some current professional society statements [13,14] would support offering expanded cfDNA to families with known t(11;22) with appropriate counseling and the provision that diagnostic testing before or after birth is needed to confirm any result, but particularly a positive result. Prenatal cfDNA screening is available as early as nine weeks’ gestation; performing prenatal cfDNA screening at the end of the first trimester would still allow time for CVS if desired, although there is a lack of data in regard to placental mosaicism for this finding. However, one study suggests that experience from cytogenetic testing of cytotrophoblast of chorionic villi is reliable in families carrying translocations, which would similarly imply mosaicism is unlikely to impact prenatal cfDNA screening [25].

Consistent with the previous literature, cases affected with ES in this cohort were primarily inherited from maternal translocations [2,4,7,9]. There was one case with an abnormal cfDNA result suggesting an atypical segregation, likely adjacent-2 segregation with der(11), that was from a known paternal translocation, but did not have confirmatory diagnostic testing. This case ended in a pregnancy loss at the end of the first trimester; other abnormal translocation byproducts from t(11;22) are associated with early pregnancy loss, so this outcome would be consistent with the literature. Known maternal cases had an almost equal split between positive/likely positive outcomes and true negative outcomes. All confirmed paternal cases were negative except the atypical segregation case. For cases affected with ES where the parental translocation carrier is unknown, starting with maternal karyotype before paternal testing is reasonable based on this study and the existing literature [2,3,5,7,10]. If the maternal karyotype is negative, the paternal karyotype is an appropriate next step.

Screening for smaller CNVs by prenatal cfDNA is, by nature, more challenging than screening for larger CNVs. In all positive ES cases in the cohort, the larger 11q event was detected and reported in accordance with assay validation; however, screening for the smaller 22q event is more challenging. Although the assays are not validated to report duplications in the region <7 Mb, the 22q duplication was noted as a courtesy in the lab director comments if detected. For two cases, the 22q duplication was not noted on the initial cfDNA results; for one case, the 22q duplication was seen in the retrospective data review. But, for the other case, even with a priori knowledge of the duplication (confirmed on diagnostic testing), there were no aberrations noted on the cfDNA sequencing data for 22q. This raises an important limitation of the technology for these cases as sensitivity for the smaller CNVs that can result from the translocation is unknown and is undoubtedly lower than the detection of the larger unbalanced byproducts. The genome-wide cfDNA assay performed at this laboratory utilizes an increased number of sequencing reads as compared to the ‘traditional’ assay, which inherently will improve the detection of CNVs. Given this increased read depth and the fact that this assay is validated to report large CNVs of at least 7 Mb (which would cover the 11q CNVs seen with the translocation), the genome-wide assay is likely the most appropriate *screening* option for these families, although diagnostic testing via amniocentesis or CVS remains the gold standard for prenatal diagnosis of ES and other chromosome abnormalities in pregnancy. When ‘traditional’ cfDNA was ordered but the provider requested a verbal review of the data, this was provided as a courtesy. However, the laboratory also recommended repeat testing using genome-wide cfDNA or diagnostic testing.

There were two cases with screen positive results on cfDNA that are presumably unrelated to the familial translocations (both maternal), with one case of 45,X and one case of apparent full trisomy 22. Without diagnostic testing in these cases, our conclusions around these events are limited. However, there is no reason to believe that presence of t(11;22) in any form (balanced or unbalanced) would impact the ability of the assay to screen the rest of the genome for chromosome abnormalities and test performance would be expected to be within the specifications of the validation studies [16,17,20,21,26]. Theoretically, the trisomy 22 result could be related to another atypical, unbalanced byproduct of t(11;22) since cfDNA screening cannot ascertain structure; however, the cfDNA sequencing data appeared to be consistent with other confirmed, non-mosaic trisomy 22 cases.

Although the details on ultrasound findings in our study are limited and undeniably incomplete, the details, where known, were provided to add to the existing (limited) literature on the prenatal phenotype of this syndrome. As reportedly previously, diaphragmatic hernia, Dandy–Walker syndrome, microretrognathia, and growth restriction were common findings [2,8,11,23,27,28]. Facial features may become coarser over time, although microretroghanthia may become less noticeable [2,5].

### 4.2. Limitations

This study is primarily limited by the fact that it is a retrospective cohort based on data provided to a commercial laboratory. Therefore, the data are certainly incomplete. Cases would only be identified for inclusion if the provider noted a translocation on the test request form (TRF) or in other communications with the laboratory such that it was noted in the clinical database, or if the case had a screen positive result consistent with ES. Although no ‘false negative’ cases were reported to the lab, cases with ES that were missed by cfDNA and were not reported to have a family history of t(11;22) would not have been flagged by the search. Lastly, cases with only one of the two events, especially a 22q-only event, may not have been ascertained by the search process. Clinical details, including diagnostic testing and pregnancy outcome, were not available for all cases, although efforts were made to contact the providers for outcome information. In some cases, the patient was lost to follow-up by the practice; in other cases, we were unable to reach the ordering provider. Lastly, most screen-negative cases did not have confirmatory testing and normal birth outcomes were used to determine a ‘true negative’ categorization. It is possible that a phenotypically normal neonate may have ES or another chromosome abnormality (either related to t(11;22) or unrelated) not identified by the screening. A lack of karyotype in screen-negative cases also limits our ability to draw conclusions around the distribution of balanced versus normal chromosome outcomes for those cases.

Although not observed in this study, false positive results are possibly related to the typical technical or biological limitations of prenatal cfDNA screening, such as cotwin demise of an affected fetus, laboratory sequencing error or artifact, maternal health conditions, or an isolated CNV in one of the target regions that turns out to be a false positive. As discussed above, CPM for unbalanced translocations in known translocation carriers is unlikely based on one study [25], but is hypothetically possible.

This study is also limited by a significant ascertainment bias since all cases were pregnancies that were ongoing into the late first trimester. As captured in the outcomes, a handful of cases ended in a pregnancy loss shortly after the prenatal cfDNA screening. Furthermore, the handful of cases with PGT-SR/IVF with transfer of a presumably unaffected embryo significantly lowers the a priori risk for those cases.

### 4.3. Conclusions and Future Directions

This study shows that prenatal cfDNA screening is likely to be a reasonable option for families with a history of t(11;22), with preference given to genome-wide cfDNA screening. Families can gain reassurance from screen negative results, and screen positive results are likely to be confirmed by diagnostic testing. The only existing alternative option for screening is ultrasound. However, more studies are needed to elaborate on test performance, which remains challenging given the prevalence of the condition. Diagnostic testing remains the gold standard for the prenatal diagnosis of ES and other fetal chromosome conditions and is recommended before any irreversible decisions are made.

Additionally, a future study could explore the clinical utility and changes to pregnancy and neonatal management based on a prenatal or early neonatal diagnosis prompted by high-risk prenatal cfDNA screening results. Given the complex medical issues associated with this condition and the high prevalence of congenital anomalies, a prenatal diagnosis in the first trimester or early second trimester could allow for referral for specialty ultrasound, including fetal echocardiogram, growth ultrasounds throughout pregnancy, and consultation with subspecialists before birth, to allow families to coordinate care and plans for an affected child. Other families may choose to pursue alternative options, such as termination of pregnancy, placement for adoption, or palliative care.

Although the availability of a screening option is promising for these families, prenatal counseling remains challenging given the variety of possible outcomes for ES. Hopefully, continued research on the prenatal phenotypes and outcomes will provide continued data to inform those conversations with patients.

## Figures and Tables

**Figure 1 genes-14-01924-f001:**
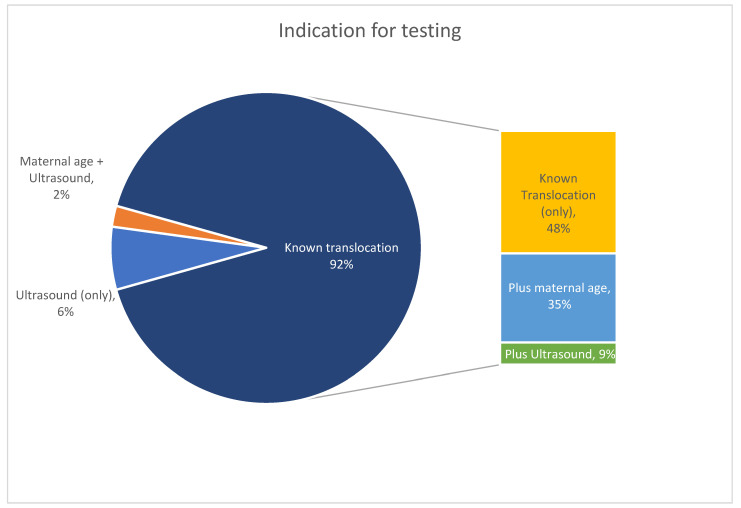
Indication for testing; the cases in the pop-out include all cases where the known translocation was part of the indication for testing, either in isolation or in combination with another indication as shown.

**Figure 2 genes-14-01924-f002:**
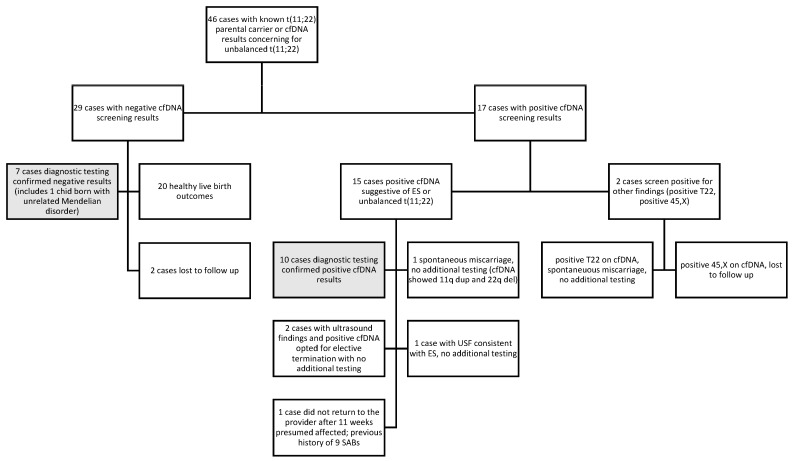
Flow chart showing the breakdown of the study cohort and diagnostic testing and pregnancy outcomes. The light gray boxes indicate the cases with confirmatory diagnostic testing. cfDNA = cell-free DNA, ES = Emanuel syndrome, SABs = spontaneous miscarriage, USF = ultrasound findings, T22 = trisomy 22.

**Figure 3 genes-14-01924-f003:**
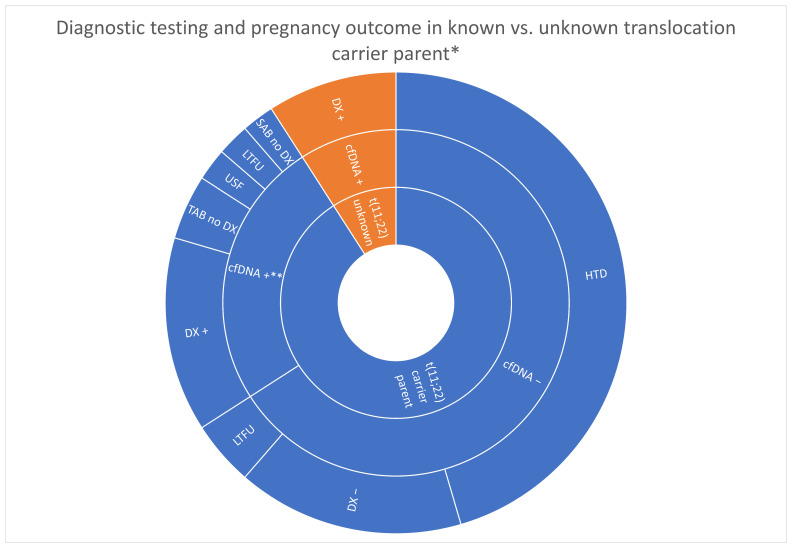
Sunburst chart showing the diagnostic testing and delivery outcomes for known t(11;22) carrier parents versus couples that were not known to have t(11;22) prior to their first cfDNA screening in the cohort. cfDNA + = cfDNA positive; DX + = diagnostic testing positive; SAB no DX = miscarriage with no diagnostic testing; TAB no DX = pregnancy termination without diagnostic testing; LTFU = lost to follow-up; USF = ultrasound findings; cfDNA− = cfDNA negative; DX− = diagnostic testing negative; HTD = unaffected birth and/or healthy, term delivery. * Unknown prior to prenatal cfDNA screening in first pregnancy in cohort, ** Excluded T22 and 45,X cfDNA positive cases.

**Table 1 genes-14-01924-t001:** Details of the 32 patients/46 pregnancies in the study cohort, with details about cfDNA testing, pregnancy outcome, and diagnostic testing results.

Patient #	Case	GA	cfDNA Indication	Known Translocation? (Origin)	cfDNA Assay Type	cfDNA Result (FF)	Diagnostic Testing + Pregnancy Outcome	Category
1 *	-	9.00	AMA, Tx Hx	Yes (Paternal)	Genome-wide	Negative (8.57%)	CVS karyotype: one twin WNL and one twin balanced carrier	Concordant negative
2	A	10.14	Tx Hx	Yes (Paternal)	Genome-wide	Positive (11 and 22) (9.94%)	SAB; No testing.	Likely concordant positive
	B	10.86	Tx Hx	Yes (Paternal)	Genome-wide	Negative (7.48%)	Healthy, term delivery. No testing.	Concordant negative
3	-	9.00	Tx Hx	Yes (Maternal)	Genome-wide	Negative (12.51%)	CVS karyotype: WNL	Concordant negative
4	-	9.00	Tx Hx	Yes (Paternal)	‘Traditional’	Negative (verbal requested) (6.68%)	Postnatal karyotype: balanced carrier	Concordant negative
5 *	-	10.00	Tx Hx	Yes (Maternal)	‘Traditional’	Negative (verbal requested) (6.57%)	Healthy, term delivery. No testing.	Concordant negative
6	-	11.57	Tx Hx	Yes (Maternal)	Genome-wide	Positive (11 and 22) (6.50%)	CVS karyotype and array: 47,XY,+der(22)t(11;22)(q23.3;q11.21); 18.26 Mb terminal duplication of 11q23.3->q25; 3.84 Mb proximal duplication of 22q11.1->q11.21	Concordant positive
7	-	11.71	AMA, Tx Hx	Yes (Maternal)	Genome-wide	Negative (7.42%)	LTFU	LTFU
8	-	12.14	Tx Hx	Yes (Maternal)	Genome-wide	Positive (11 only) (4.54%)	Amniocentesis karyotype: 47,XY,+der(22)t(11;22)(q23.3;q11.2) Patient is gravida 11—all early SABs	Concordant positive
9 *	A	10.14	AMA, Tx Hx	Yes (Maternal)	Genome-wide	Positive (11 and 22) (15.27%)	Multiple anomalies; TAB; No testing.	Likely concordant positive
	B	9.00	AMA, Tx Hx	Yes (Maternal)	Genome-wide	Negative (9.46%)	Egg donor/IVF; Monochorionic/diamniotic twins; CVS karyotype: WNL	Concordant negative
10	-	11.71	Tx Hx	Yes (Maternal)	Genome-wide	Negative (10.91%)	Healthy, term delivery. No testing.	Concordant negative
11	A	10.00	AMA, Tx Hx	Yes (Paternal)	Genome-wide	Negative (10.08%)	PGT-SR with presumably balanced/normal embryo; Healthy, term delivery. No testing	Concordant negative
	B	9.71	AMA, Tx Hx	Yes (Paternal)	Genome-wide	Negative (10.95%)	Amniocentesis karyotype and array: balanced carrier	Concordant negative
12	-	21.57	USF	No	Genome-wide	Positive (11 and 22) (7.04%)	Diaphragmatic hernia on ultrasound. Previous pregnancy also had diaphragmatic hernia, neonatal death with no testing. Postnatal array: 18.2 Mb terminal duplication of 11q23.3->q25; 3.4 Mb proximal duplication of 22q11.1->q11.21	Concordant positive
13	A	9.00	AMA, Tx Hx	Yes (Maternal)	Genome-wide	Negative (15.75%)	Healthy, term delivery. No testing. Reports previous child with ES.	Concordant negative
	B	10.29	AMA, Tx Hx	Yes (Maternal)	Genome-wide	Negative (10.76%)	Healthy, term delivery. No testing.	Concordant negative
14	A	13.57	USF, Tx Hx	Yes (Maternal)	Genome-wide	Positive (11 and 22) (10.81%)	Maternal family history of ES with history of 2 prior SABs; Maternal karyotype concurrent to cfDNA confirmed carrier status; Amniocentesis: FISH showed 3 copies of 11q23 and 22q11.2; Products of conception karyotype: 47,XX,+der(22)t(11;22)(q23.3;q11.2); TAB	Concordant positive
	B	11.00	Tx Hx	Yes (Maternal)	Genome-wide	Negative (6.31%)	Healthy, term delivery. No testing.	Concordant negative
15	A	12.00	Tx Hx	Yes (Maternal)	Genome-wide	Negative (10.20%)	Healthy, term delivery. No testing.	Concordant negative
	B	12.29	Tx Hx	Yes (Maternal)	Genome-wide	Positive (11 and 22) (9.95%)	SAB; CVS karyotype: 47,XX,+der(22)t(11;22)(q23.3;q11.2)	Concordant positive
	C	9.00	Tx Hx	Yes (Maternal)	Genome-wide	Negative (7.79%)	CVS karyotype: balanced carrier	Concordant negative
	D	10.71	USF, Tx Hx	Yes (Maternal)	Genome-wide	Positive (11 only) (8.33%)	Patient did not return for care; History of 9+ SABs and reports previous children affected; 22q dup noted on data review	Likely concordant positive
16	A	12.43	AMA, Tx Hx	Yes (Paternal)	Genome-wide	Negative (10.51%)	Healthy, term delivery. No testing.	Concordant negative
	B	10.14	AMA, Tx Hx	Yes (Paternal)	Genome-wide	Negative (7.10%)	Healthy, term delivery. No testing.	Concordant negative
17	-	26.14	USF	No	Genome-wide	Positive (11 and 22) (15.41%)	Dandy–Walker malformation on ultrasound; Amniocentesis karyotype: 47,XX,+der(22)t(11;22)(q23.3;q11.2). Delivered affected infant; no additional outcome available.	Concordant positive
18	A	12.29	AMA, Tx Hx	Yes (Maternal)	Genome-wide	Negative (14.35%)	Healthy, term delivery. No testing.	Concordant negative
	B	12.00	AMA, Tx Hx	Yes (Maternal)	Genome-wide	Negative (13.76%)	Healthy, term delivery. No testing.	Concordant negative
19	-	12.86	Tx Hx	Yes (Maternal)	Genome-wide	Negative (11.43%)	Healthy, term delivery. No testing.	Concordant negative
20	-	19.86	AMA, USF	No	Genome-wide	Positive (11 and 22) (9.76%)	Diaphragmatic hernia and mild ventriculomegaly on ultrasound; Amniocentesis array: 18.3 Mb duplication of 11q23.3->11qter; 3.42 terminal duplication of 22qter->22q11.21	Concordant positive
21	-	11.43	Tx Hx	Yes (Maternal)	Genome-wide	Negative (11.24%)	LTFU	LTFU
22	-	23.14	USF, Tx Hx	Yes (Maternal)	Genome-wide	Positive (11 and 22) (7.54%)	Dandy–Walker malformation on ultrasound; history of several SABs and one healthy child; Postnatal karyotype: 47,XY,+der(22)t(11;22)(q23.3;q11.2)	Concordant positive
23	-	11.00	Tx Hx	Yes (Paternal)	Genome-wide	Negative (6.46%)	Healthy, term delivery. No testing.	Concordant negative
24	A	22.29	USF	No	‘Traditional’	Positive (verbal requested) (14.35%)	Multiple anomalies including micrognathia, growth restriction, and possible heart defect. Amniocentesis karyotype showed ‘derivative chromosome 22′ per verbal report from genetic counselor	Concordant positive
	B	9.29	Tx Hx	Yes (Maternal)	Genome-wide	Positive (11 and 22) (8.49%)	Multiple anomalies including micrognathia, growth restriction, and possible heart defect; No testing; TAB	Likely concordant positive
	C	9.14	Tx Hx	Yes (Maternal)	Genome-wide	Negative (12.24%)	PGT-SR, Anomalies/complication related to Mendelian disorder diagnosed via WES. Amniocentesis karyotype: WNL.	Concordant negative
25	-	11.57	Tx Hx	Yes (Maternal)	Genome-wide	Negative (5.50%)	Healthy, term delivery. No testing. Previous child with ES.	Concordant negative
26	A	10.00	AMA, Tx Hx	Yes (Maternal)	Genome-wide	Positive (trisomy 22) (5.88%)	SAB at 11 weeks; No testing.	Likely concordant positive
	B	10.43	AMA, Tx Hx	Yes (Maternal)	Genome-wide	Negative (5.94%)	Healthy, term delivery. No testing.	Concordant negative
	C	10.29	AMA, Tx Hx	Yes (Maternal)	Genome-wide	Negative (5.57%)	Healthy, term delivery. No testing.	Concordant negative
27	-	10.86	Tx Hx	Yes (Maternal)	Genome-wide	Negative (11.24%)	Healthy, term delivery. No testing. Previous child with another chromosome disorder which prompted maternal karyotype.	Concordant negative
28	-	11.57	Tx Hx	Yes (Maternal)	Genome-wide	Negative (6.53%)	Healthy, term delivery. No testing.	Concordant negative
29	-	18.86	USF, Tx Hx	Yes (Maternal)	Genome-wide	Positive (11 and 22) (7.43%)	Multiple anomalies including micrognathia, diaphragmatic hernia, Dandy–Walker malformation, and two vessel cord; Amniocentesis karyotype and array: 47,XX,+der(22)t(11;22)(q23; q11.2); TAB	Concordant positive
30	-	12.14	Tx Hx	Yes (Maternal)	Genome-wide	Positive (11 and 22) (6.83%)	Ultrasound anomalies including recessed chin and growth restriction; ‘partial T11′ in previous pregnancy	Likely concordant positive
31	-	12.00	AMA, Tx Hx	Yes (Maternal)	‘Traditional’	Negative (Verbal requested) (10.03%)	PGT-SR: carrier male; healthy, term delivery. No additional testing.	Concordant negative
32	-	10.00	Tx Hx	Yes (Maternal)	Genome-wide	Positive (45,X) (5.09%)	Normal nuchal translucency on ultrasound; LTFU.	LTFU

* indicates twin pregnancy, GA = gestational age, FF = fetal fraction, AMA = Advanced maternal age 35+, Tx Hx = Known history of translocation, USF = Ultrasound findings, DCVS = chorionic villus sampling, SAB = spontaneous abortion, LTFU = lost to follow-up, WNL = within normal limits, Mb = Megabase, PGT-SR = Preimplantation genetic testing for structural rearrangements, WES = Whole exome sequencing, TAB = elective termination, ES = Emanuel syndrome, T11 = Trisomy 11.

**Table 2 genes-14-01924-t002:** Details of the positive cfDNA screening results in the study cohort.

Study Identifier	cfDNA Positive Reported	Z-Scores	Details of cfDNA Findings
Patient 2-A	Positive (11 and 22)	11p: 45.3 22q: −4.88	116.65 Mb gain of 11p15.5-q23.3 2.75 Mb loss of 22q11.2-q11.2 (DiGeorge region)
Patient 6	Positive (11 and verbal 22)	11q: 11.1 22q: 5.23	18.35 Mb gain of 11q23.3-q25 Verbal report of small ~3 Mb dup on 22q
Patient 8	Positive (11)	11q: 8.86	20.6 Mb gain of 11q23.2-q25
Patient 9-A	Positive (11 and 22)	11q: 33.16 22q: 8.76	18.25 Mb gain of 11q23.3-q25 3.0 Mb gain of 22q11.1-q11.21
Patient 12	Positive (11 and 22)	11q: 15.34 22q: 6.37	18.1 Mb gain of 11q23.3-q25 2.85 Mb gain of 22q11.1-q11.21
Patient 14-A	Positive (11 and 22)	11q: 22.82 22q: 6.89	18.40 Mb gain of 11q23.3-q25 3.0 Mb gain of 22q11.1-q11.21
Patient 15-B	Positive (11 and 22)	11q: 18.27 22q: 6.34	18.5 Mb gain of 11q23.3-q25 2.8 Mb gain of 22q11.1-q11.2
Patient 15-D	Positive (11)	11q: 23.56 22q: 8.69	18.3 Mb gain of 11q23.3-q25 2.05 Mb gain of 22q11.21-q11.21 noted on retrospective data review
Patient 17	Positive (11 and 22)	11q: 37.21 22q: 14.32	18.25 Mb gain of 11q23.3-q25 3.0 Mb gain of 22q11.1-q11.21
Patient 20	Positive (11 and 22)	11q: 19.92 22q: 7.77	18.3 Mb gain of 11q23.3-q25 3.55 Mb gain of 22q11.1-q11.21
Patient 22	Positive (11 and 22)	11q: 13.76 22q: 4.54	18.3 Mb gain of 11q23.3-q25 3.65 Mb gain of 22q11.1-q11.21
Patient 24-A	Positive-verbal review requested of 11/22	11q: 22.18 22q: 6.79	Verbal report of duplications on 11q and 22q in cfDNA sequencing data
Patient 24 B	Positive (11 and 22)	11q: 14.22 22q: 7.68	17.75 Mb gain of 11q23.3-q25 4.75 Mb gain of 22q11.1-q11.21
Patient 29	Positive (11 and 22)	11q: 20.91 22q: 8.00	18.25 Mb gain of 11q23.3-q25 3.55 Mb gain of 22q11.1-q11.21
Patient 30	Positive (11 and 22)	11q: 14.25 22q: 4.65	18.05 Mb gain of 11q23.3-q25 3.0 Mb gain of 22q11.1-q11.21
Patient 32	Positive (45,X)	NA	cfDNA data consistent with monosomy X Chromosome 11 and 22 data unremarkable
Patient 26-A	Positive (T22)	NA	cfDNA data consistent with full trisomy 22 Chromosome 11 data unremarkable

## Data Availability

All relevant data are included in the manuscript.

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
