# Peer review of "Prenatal cfDNA Screening for Emanuel Syndrome and Other Unbalanced Products of Conception in Carriers of the Recurrent Balanced Translocation t(11;22): One Laboratory’s Retrospective Experience"

_genes, 2023, doi:10.3390/genes14101924_

Round 1
Reviewer 1 Report
General Comments:
1. The manuscript is very well organized and written.
2. The title should reflect the retrospective nature of the data analysis as mentioned by authors on page 15; line 362
3. There is some redundancy in repeating the same sentence in different paragraphs. it is recommended that need to be checked and edited to make the manuscript more concise.
Specific comments:
1. Page 2 line 44. if de novo origin of translocations (11;220 are known to exist, this need to be mentioned
2. Page 3: For follow up confirmation of cfDNA screening Tier 1 CMA test is recommended. Information on CMA testing of the cohort need to be added
3. Page 4, line 158. Mention the cohort size N=? included to filter the ES cases
4. Page 13, line 297-299, inlclude a reference for the statement
5. Page 14, line 324, add a sentence all negative cases of maternal chromosome analysis reflex to paternal chromosome analysis.
6. Page 14, line 325-344. WGS is still not a Tier 1 test for CNVs and need to be clearly stated and CMA need to be mentioned as the Tier 1 for CNVs until practice guidelines are updated.
Author Response
Dear Authors,
you can read my revision on the side of your manuscript submitted.
- In the Introduction, could be better to give a major description of the history of the syndrome (why it was named Emanuel?)
- We thank the reviewer for this comment. We have added a section to the introduction to address this concern.
- On the second page, the Authors describe the maternal prevalence of the hereditary syndrome, please try to give an explanation by the literature and in the discussion, give your hypothesis
- We thank the reviewer for this comment. We have added a section to the introduction to address this concern.
- Ever on the second page, could be better to give a major description of the symptoms
- We thank the reviewer for this comment. We have added a section to the introduction to address this concern.
- To the end of the second page there is a description of the utility of this genetic test, however in my opinion it is important to improve this sentence because there is an ethical conflict in the description of the Authors. If they wish a clinical utility of the test, they have also to mention the use of your prenatal screening to cure the fetus affected, for example cardiovascular, gastrointestinal, orthopaedic malformation, deafness...
- We thank the reviewer for this comment. To the reviewer’s point, this study did not assess the changes in pregnancy management based on the cfDNA result, therefore, we agree that it is not appropriate to discuss clinical utility. However, we did add a paragraph to the ‘future directions’ section of the discussion to hypothesize about ways this testing could impact clinical utility and to highlight this as an area for a future study.
- On page 15, regarding to the micrognatia, it became less apparent with age. See OMIM
- We thank the reviewer for this comment and have added another sentence to this section of the discussion.
Reviewer 2 Report
Dear Authors,
you can read my revision on the side of your manuscript submitted.
1) In the Introduction, could be better to give a major description of the history of the syndrome (why it was named Emanuel?) 2) On the second page, the Authors describe the maternal prevalence of the hereditary syndrome, please try to give an explanation by the literature and in the discussion, give your hypothesis 3) Ever on the second page, could be better to give a major description of the symptoms 4) To the end of the second page there is a description of the utility of this genetic test, however in my opinion it is important to improve this sentence because there is an ethical conflict in the description of the Authors. If they wish a clinical utility of the test, they have also to mention the use of your prenatal screening to cure the fetus affected, for example cardiovascular, gastrointestinal, orthopaedic malformation, deafness... 5) On page 15, regarding to the micrognatia, it became less apparent with age. See OMIM

Author Response
The authors present a significant and timely study regarding the potential clinical applicability of cell-free fetal DNA (cfDNA) screening for the identification of Emanuel syndrome (ES), resulting from an unbalanced translocation between chromosomes 11 and 22. The authors provide a retrospective study cohort comprising 46 cases that were received by the commercial laboratory between 2013 to 2023. Inclusion criteria encompassed screen-positive cases characterised by complex CNVs on chromosomes 11 and 22, screen-negative cases that were tested due to a known familial or the 11;22 translocation, and additional pregnancies sent for cfDNA screening at the same laboratory for the same patient, which were reviewed to confirm parental carrier status of t(11;22). The authors reported no discordant results, revealing ten true positives confirmed through invasive prenatal diagnostic testing, six likely concordant positives based on known translocations and consistent cfDNA data, and 26 true negatives.
The importance of this study should be highlighted and I believe the study would significantly add to the body of literature. However, I would like to describe here a few of my concerns:
We thank the reviewer for this comment and the positive feedback.
Page 2, line 57: I would like to recommend that the authors consider incorporating the updated reference: Emanuel BS. 2017. Emanuel syndrome. GeneReviews. [Internet]. Seattle (WA): University of Washington, Seattle; 1993. 2007 Apr 20 (updated 2017 Aug 31).
We thank the reviewer for this comment and have made the suggested revision.
Page 3, Materials & Methods section: The authors omitted details concerning fetal fractions, which could prove beneficial to the readers of the journal. Providing information on the minimum fetal fraction necessary for genome-wide cfDNA screening, the rate of blood redraws, or sequencing failures would enhance comprehension. Furthermore, the authors might consider including data on the minimum acceptable fetal fraction for pregnancies yielding negative test results, as well as the fetal fraction in the context of twin pregnancies. Additionally, if applicable, they should consider indicating z-scores and sequencing read depths
We thank the reviewer for this comment. We have added sentences to the methods to include reference to fetal fraction methodology/thresholds and to cite published non-reportable rates for both assays since no-call rates were not assessed by the study design in this cohort. Details around specific sequencing read depths for each assay are considered proprietary by the company and not disclosed.
Page 3 line 140: It is noted that not all positive NIPS cases that underwent invasive prenatal diagnosis were subject to verification through the three distinct methods, namely conventional karyotyping, FISH and CMA. Perhaps the authors may want to include this in the study limitations.
We thank the reviewer for this comment. We acknowledge in the limitations section that we did not have prenatal diagnostic results (or birth outcomes) for all cases: “Clinical details, including diagnostic testing and pregnancy outcome, were not available for all cases, although efforts were made to contact the providers for outcome in-formation. In some cases, the patient was lost to follow-up by the practice; in other cases, we were unable to reach the ordering provider. Lastly, most screen negative cases did not have confirmatory testing and normal birth outcomes were used to determine a ‘true negative’ categorization. It is possible that a phenotypically normal neonate may have ES, or another chromosome abnormality (either related to t(11;22) or unrelated) not identified by the screening. Lack of karyotype in screen negative cases also limits our ability to draw conclusions around the distribution of balanced versus normal chromosome outcomes for those cases.”
Page 4, table 1: Recommend that the authors consider including information on fetal fractions and z-scores.
We thank the reviewer for this comment. We have included fetal fractions in this table and we provided Z-scores in Table 2 for the screen positive results.
Page 4, table 1: Recommend that the authors consider including information of twin pregnancy chorionicity and zygosity for added clarity.
We thank the reviewer for this comment; unfortunately, the laboratory does not routinely receive this clinical information and thus it is not available for the cases or included in the clinical data in the study.
Page 15, line 354: The authors may want to include information from Srebniak et al., 2018 that the likelihood of inherited, unbalanced translocations exhibiting mosaicism and absent from the cytotrophoblast is low. Therefore, confined placental mosaicism is unlikely to impact the accuracy of NIPS results.
We thank the reviewer for this comment and have incorporated this information and reference into our discussion.
Page 15, line 379: While the study did not report any false positive results, it would be valuable for the authors to consider presenting a hypothetical scenario in which a false positive result could occur, along with potential strategies to mitigate the limitations associated with false positive reports.
We thank the reviewer for this comment and have incorporated this into our discussion section.
Page 15, lines 388 go 390: The authors concluded that “Families can gain reassurance from screen negative results, and screen positive results are highly likely to be confirmed by diagnostic testing. Clearly, it is a better screening option than any existing alternatives.” As there are only a limited number of studies on ES screening using cfDNA studies to date, it would be advisable for the authors to temper these statements until a more substantial body of research and literature becomes available.
We thank the reviewers for this comment and have revised that paragraph with the reviewer’s feedback in mind.
In conclusion, I find that the manuscript is well-written, and the significant findings and points have been presented with clarity. I am pleased to recommend that Genes accept the manuscript for publication, pending satisfactory resolution of the concerns I have raised above.
We thank the reviewer for this comment.
Reference:
Srebniak MI, Vogel I, Van Opstal D. Is carriership of a balanced translocation or inversion an indication for noninvasive prenatal testing?. Expert Rev Mol Diagn. 2018;18:1–3.
Reviewer 3 Report
The authors present a significant and timely study regarding the potential clinical applicability of cell-free fetal DNA (cfDNA) screening for the identification of Emanuel syndrome (ES), resulting from an unbalanced translocation between chromosomes 11 and 22. The authors provide a retrospective study cohort comprising 46 cases that were received by the commercial laboratory between 2013 to 2023. Inclusion criteria encompassed screen-positive cases characterised by complex CNVs on chromosomes 11 and 22, screen-negative cases that were tested due to a known familial or the 11;22 translocation, and additional pregnancies sent for cfDNA screening at the same laboratory for the same patient, which were reviewed to confirm parental carrier status of t(11;22). The authors reported no discordant results, revealing ten true positives confirmed through invasive prenatal diagnostic testing, six likely concordant positives based on known translocations and consistent cfDNA data, and 26 true negatives.
The importance of this study should be highlighted and I believe the study would significantly add to the body of literature. However, I would like to describe here a few of my concerns:
Page 2, line 57: I would like to recommend that the authors consider incorporating the updated reference: Emanuel BS. 2017. Emanuel syndrome. GeneReviews. [Internet]. Seattle (WA): University of Washington, Seattle; 1993. 2007 Apr 20 (updated 2017 Aug 31).
Page 3, Materials & Methods section: The authors omitted details concerning fetal fractions, which could prove beneficial to the readers of the journal. Providing information on the minimum fetal fraction necessary for genome-wide cfDNA screening, the rate of blood redraws, or sequencing failures would enhance comprehension. Furthermore, the authors might consider including data on the minimum acceptable fetal fraction for pregnancies yielding negative test results, as well as the fetal fraction in the context of twin pregnancies. Additionally, if applicable, they should consider indicating z-scores and sequencing read depths.
Page 3 line 140: It is noted that not all positive NIPS cases that underwent invasive prenatal diagnosis were subject to verification through the three distinct methods, namely conventional karyotyping, FISH and CMA. Perhaps the authors may want to include this in the study limitations.
Page 4, table 1: Recommend that the authors consider including information on fetal fractions and z-scores.
Page 4, table 1: Recommend that the authors consider including information of twin pregnancy chorionicity and zygosity for added clarity.
Page 15, line 354: The authors may want to include information from Srebniak et al., 2018 that the likelihood of inherited, unbalanced translocations exhibiting mosaicism and absent from the cytotrophoblast is low. Therefore, confined placental mosaicism is unlikely to impact the accuracy of NIPS results.
Page 15, line 379: While the study did not report any false positive results, it would be valuable for the authors to consider presenting a hypothetical scenario in which a false positive result could occur, along with potential strategies to mitigate the limitations associated with false positive reports.
Page 15, lines 388 go 390: The authors concluded that “Families can gain reassurance from screen negative results, and screen positive results are highly likely to be confirmed by diagnostic testing. Clearly, it is a better screening option than any existing alternatives.” As there are only a limited number of studies on ES screening using cfDNA studies to date, it would be advisable for the authors to temper these statements until a more substantial body of research and literature becomes available.
In conclusion, I find that the manuscript is well-written, and the significant findings and points have been presented with clarity. I am pleased to recommend that Genes accept the manuscript for publication, pending satisfactory resolution of the concerns I have raised above.
Reference:
Srebniak MI, Vogel I, Van Opstal D. Is carriership of a balanced translocation or inversion an indication for noninvasive prenatal testing?. Expert Rev Mol Diagn. 2018;18:1–3.
Author Response
General Comments:
- The manuscript is very well organized and written.
- We thank the reviewer for this comment.
- The title should reflect the retrospective nature of the data analysis as mentioned by authors on page 15; line 362
- We thank the reviewer for this comment and have added the word retrospective to our title. We kindly request that the editors help us to update the manuscript title in the editorial system, as well.
- There is some redundancy in repeating the same sentence in different paragraphs. it is recommended that need to be checked and edited to make the manuscript more concise.
- We thank the reviewers for this comment. We have reviewed the manuscript and made modifications based on this comment.
Specific comments:
- Page 2 line 44. if de novo origin of translocations (11;220 are known to exist, this need to be mentioned
- We thank the reviewer for this comment and have added a sentence to address this point to the introduction.
- Page 3: For follow up confirmation of cfDNA screening Tier 1 CMA test is recommended. Information on CMA testing of the cohort need to be added
- We thank the reviewer for this comment. Some professional societies do not recommend a specific diagnostic test for confirmation of cfDNA, just that diagnostic testing should be offered to confirm results. Therefore, we left our language in the manuscript more general, to be inclusive of all society guidelines: “Diagnostic testing would be needed to definitively confirm or rule out a diagnosis of a chromosome abnormality in a pregnancy.”
- Furthermore, in this cohort, as most of the diagnostic tests were performed by the ordering provider and relayed to the laboratory, we do not have access to the details regarding the type of array platform/details about the array that were used for each patient. Lastly, some patients had confirmation by karyotype only.
- Page 4, line 158. Mention the cohort size N=? included to filter the ES cases
- We thank the reviewer for this comment, and have added a note that the database search included over 2 million prenatal cfDNA samples.
- Page 13, line 297-299, inlclude a reference for the statement
- We thank the reviewer for this comment and have added a reference as directed.
- Page 14, line 324, add a sentence all negative cases of maternal chromosome analysis reflex to paternal chromosome analysis.
- We thank the reviewer for this comment and have added a statement as suggested.
- Page 14, line 325-344. WGS is still not a Tier 1 test for CNVs and need to be clearly stated and CMA need to be mentioned as the Tier 1 for CNVs until practice guidelines are updated.
- We thank the reviewer for this comment and have added a statement reiterating that prenatal diagnosis is the gold standard for detection of chromosome abnormalities in pregnancy.